# A Customizable Analysis Flow in Integrative Multi-Omics

**DOI:** 10.3390/biom10121606

**Published:** 2020-11-27

**Authors:** Samuel M. Lancaster, Akshay Sanghi, Si Wu, Michael P. Snyder

**Affiliations:** 1Department of Genetics, Stanford School of Medicine, Stanford, CA 94305, USA; 2Cardiovascular Institute, Stanford School of Medicine, Stanford, CA 94305, USA

**Keywords:** multi-omics, multi-omics analysis, study design, bioinformatics, machine learning, analysis flow

## Abstract

The number of researchers using multi-omics is growing. Though still expensive, every year it is cheaper to perform multi-omic studies, often exponentially so. In addition to its increasing accessibility, multi-omics reveals a view of systems biology to an unprecedented depth. Thus, multi-omics can be used to answer a broad range of biological questions in finer resolution than previous methods. We used six omic measurements—four nucleic acid (i.e., genomic, epigenomic, transcriptomics, and metagenomic) and two mass spectrometry (proteomics and metabolomics) based—to highlight an analysis workflow on this type of data, which is often vast. This workflow is not exhaustive of all the omic measurements or analysis methods, but it will provide an experienced or even a novice multi-omic researcher with the tools necessary to analyze their data. This review begins with analyzing a single ome and study design, and then synthesizes best practices in data integration techniques that include machine learning. Furthermore, we delineate methods to validate findings from multi-omic integration. Ultimately, multi-omic integration offers a window into the complexity of molecular interactions and a comprehensive view of systems biology.

## 1. Introduction

Omics measurements are unbiased samples of molecules from a biological specimen. The genome was the first ome studied [1,2], and subsequent omes followed, building off DNA sequencing technology. Transcriptomics sequences the RNA content in cells, and metagenomics sequences all the genetic material from a group of organisms, usually microbial populations. Chromatin accessibility measurements select for sections of DNA to sequence that are differentially bound by chromatin—believed to affect transcription.

Omic measurements are not limited to nucleic acid sequencing. The most common omics methods orthologous to nucleotide sequencing involve mass spectrometry (MS). These include proteomics, metabolomics, and lipidomics, which are all vitally important and account for innumerable actionable discoveries. There are many other omic measurements, which all work together to improve understanding of systems biology.

Understanding each of these omes is vitally important and integrating them provides a more comprehensive picture of biology. For example, to understand the biochemical effects of changes in transcription, one must understand the metabolome and proteome as well. However, with the different natures of omic measurements, and the fact that they are best modeled by different statistical distributions, integrating this vast information in these distinct biological layers is challenging to non-experts. Using these omic measurements as examples, we will highlight potential integration methods that will reveal trends in multi-omics data.

## 2. Analysis of Single Omics Prior to Integration

Each of these omic methods is analyzed differently, with similar analyses shared between the more similar methods. One cannot discuss multi-omic integration without first having a shared understanding of how to analyze the individual omic measurements.

### 2.1. Genome Analysis

The genome is the core ome, and it codes for the basic information that inevitably is pushed into the other omes. For example, the transcriptome is aligned with the genome. This task is complicated because of the numerous mRNA isoforms, and the non-normal distribution of reads, which can be modeled using a negative binomial distribution [3]. After alignment and normalization, the read depth is used as a measurement of expression, reviewed below. Similarly, in the metagenome data, reads are aligned with the set of known microbiome data and read depth is assumed to be an abundance of each microorganism [4]. Chromatin accessibility measurements, such as the assay for transposase-accessible chromatin using sequencing (ATACseq), follow a similar principle. In this case read depth is a measure for how open the chromatin is.

Most genomes are sequenced on an Illumina platform, generating short reads. First, the quality of these reads must be determined, which informs one how well the sequencing was performed. Generally speaking a PHRED score of 30 is used as a threshold for keeping a read, although it may be altered depending on the needs of a study [5]. These scores are saved in FASTQ files as one of the four rows for each read, and they may be pulled out using several different programs. Another main sequencing type, long read sequencing, usually allows for the retrieval of much longer (>10,000 bp) sequencing reads (e.g., PacBio) and may be used to better capture repetitive regions or insertions or deletions, but it is often more expensive per base.

The reads that pass quality controls must be aligned with a known genome. For organisms without assembled reference genomes, which are increasingly rare, such reads must first be assembled into a genome with large contiguous chunks of DNA, or contigs (reviewed in [6]). Alignment tools such as BWA and Bowtie allow alignment of reads with a given number of mismatches, because no genome will be identical to the reference genome [7,8]. These alignments generate a sequence alignment map (SAM) file and their more compressed binary format BAM file [9]. From these files, variants between the sequenced genome and referenced genome can be determined using Samtools or other software and saved as a variance call format (VCF) file [10]. These may be DNA insertions, deletions, or nucleotide variations. From these files, biologically relevant genetic differences, for example, those that affect protein translations, may be determined. In some cases, single nucleotide polymophisms (SNPs) can be associated with known phenotypes or may even be proved causative for a disease.

### 2.2. Epigenomic Analysis

Epigenomic analysis aims to understand the functional context of the genome. For example, an animal has multiple organs with the same genome, but the genes expressed vary between organs depending on the tissue’s epigenetic state. The genome is contained within a larger chromatin context that regulates which genes have access to transcriptional machinery and which are insulated from active machinery.

Various technologies have been developed to profile the epigenetic landscape, and particularly in the last decade, next-generation technologies have been applied to comprehensively map the epigenetic patterns in mammalian species [11,12]. One of the newest technologies in epigenetic analysis is assay transposase-accessible chromatin sequencing (ATAC-seq) [13]. The benefits of the ATAC-seq are (1) it provides direct evidence of genomic positions of nucleosome-depleted chromatin, which are permissible to transcriptional machinery binding, and (2) the assay only requires 10,000–50,000 cells as input, so it is particularly useful for animal tissue and limited specimens [14].

Similarly to whole-genome sequencing, ATAC-seq data are generated on the Illumina platform, giving high resolution information of open chromatin regions throughout the entire genome. After alignment with the same genome aligners, such as Bowtie, a critical step is filtering out low-quality and insignificant reads. This especially entails removing the high fraction of mitochondrial reads, which because of their high degree of accessibility are preferentially ligated with sequencing adapters. The sequencing reads are then analyzed for their pile-ups in peaks. The general purpose of peak calling is to find regions (on the order of hundreds of base pairs) that have significantly more reads piled up compared to the background reads across the genome [15]. ATAC-seq peaks represent the functional output of the data, and peaks are used in several types of analyses [16]. One very interesting analysis is transcription factor footprinting, which predicts which transcription factors are actively bound to chromatin and where the transcription factors activate transcription, giving insights into the regulatory pathways that affect gene expression [15].

### 2.3. Transcriptome Analysis

Transcriptomic data are generated in a similar way to genome sequencing libraries, except cDNA from reverse transcription is sequenced rather than genomic DNA. Aligning these reads to a transcriptome is a more complicated problem than aligning to a genome because of RNA variants, splicing, and otherwise uneven transcription of the genome. Transcriptome alignment tools require aligners such as Bowtie or BWA but require different information to annotate the transcription of the genome. The most commonly used program for transcriptome analysis is Spliced Transcripts Alignment to a Reference (STAR) [17]. This program is what is used by The Encyclopedia of DNA Elements (ENCODE), so should be used if someone wants to directly compare their results to most other experiments [18]. A newer program that is even faster than STAR is Kallisto, which is beneficial because it reduces computational expenses for very large experiments [19]. Salmon is another reputable transcriptomic software as well, among others [20]. Any of these different software algorithms will produce useful results for your experiment that may be later used for multi-omic integration.

Once this software has been run, several metrics will be generated for every transcript in each sample, including transcripts per million (TPM) and reads per kilobase of transcript, per million mapped reads (RPKM). To begin your analysis, several steps need to be taken. One analysis program in particular is used because of its end-to-end capabilities: the R package DESeq [3]. Similar packages include edgeR and limma [21,22]. The normalized reads can then be used for downstream analyses listed below. To perform custom analyses, the data should be read into a data matrix, which is helped by a program such as the R program tximport [23]. In this way TPM or RPKM can be pulled out for every sample. These should then be corrected for batch effects, for which RNAseq is particularly sensitive. The program sva::COMBAT() from R is excellently suited for batch correction [24]. Once corrected, the data are ready for downstream data analysis as illustrated below.

The first step in differential analysis workflow is data normalization, in order to guarantee the accurate comparisons of gene expression between and/or within samples. Proper normalization is essential not only for differential analysis, but also for exploratory analysis and visualization of data. The main factors that we often need to consider during count normalization are sequencing depth, gene length, and RNA composition. There are several common normalization methods to account for the “unwanted” variates, including counts per million (CPM), TPM, reads/fragments per kilobase of exon per million reads/fragments mapped (RPKM/FPKM), and DESeq2′s median of ratio trimmed mean of M values (TMM) [3,25].

CPM, TPM, and RPKM/FPKM are the traditional normalization methods for sequencing data, but they are not suitable for differential analysis due to the fact that they only account for sequencing depth and gene length, but not RNA composition. Accounting for RNA composition is especially crucial for the scenario with a few highly differentially expressed genes between samples—big differences in the number of genes expressed between samples, which can skew the traditional types of normalization methods. It is highly recommended to account for RNA composition, especially for differential analysis [3]. Due to this, TMM normalization was developed and can be conducted in the edgeR package [22]. DESeq2 package implements the normalization method of median of ratio [3]. The DESeq2 package implements transformations by computing a variance stabilizing transformation which is roughly similar to log2 transformation of data, but also deals with the sample variability of low counts, generating vst and rlog formats of data. However, both formats are designed for applications other than differential analysis, such as sample clustering and other machine learning applications.

From these data, transcript enrichment can be performed using gene ontology (GO) or another such categorization method. GO involves assigning one or more functions to each gene based on its experimental function or categorization, and these categories are assigned to several genes. Then between the cases and controls one may see whether the GO categories are significantly enriched. DAVID bioinformatics offer a wide range of enrichment methods, including GO enrichments [26], although there are many similar GO algorithms such as GOrilla [27]. Another powerful pathway analysis and mechanism elucidation tool is ingenuity pathway analysis (IPA). It is an all-in-one, Web-based software application, enabling analysis, integration, and understanding of omics data, including gene expression, miRNA, SNP microarray, proteomics, metabolomics, etc. However, one of its downsides is that it is only commercially available. Nguyen et al. [28] systematically investigated and summarized the comprehensive pathway enrichment analysis tools, and concluded that topology-based methods outperform other methods, given the fact that topology-based methods take into account the structures of the pathways and the positions of each molecule in the biological system map. The best topology-based approaches include SPIA (signaling pathway impact analysis) and the ROntoTool R package.

Deeper analyses may be performed as well. For example, from peripheral blood mononuclear cells (PBMCs) the composition of the white blood cells may be estimated from expression of marker genes using software such as immunoStates [29], although others are effective as well [30]. These data may complement integrative analyses, integrating enrichment software from various omes.

### 2.4. Metagenomic Analysis

Metagenomic analysis also is similar to other nucleic acid omes. All the genetic material from a microbiome sample, often from stool, is sequenced. This review will discuss sequencing on an Illumina platform; however, other sequencing platforms are appropriate as well. This is all the genetic material from multiple organisms, hence the metagenome. These reads must be queried against a database, similarly to the previous methods. For example, the pipeline can query the human microbiome project database not just for different taxa, but also for biochemical pathways and even related individual genes [31]. This is important because taxa alone do not provide all the functional biological data about a microbial population. Such data provide a wealth of information about several levels of the microbiome. A fast, highly sensitive, although less specific method is querying chunks of the reads, or kmers, against a database, as used in Kraken. To aid with the problems presented by genomic flexibility in microorganisms, a kmer approach is increasingly being utilized, which requires only aligning part of the read, not the entire one [32].

These methods require very deep sequencing, and a more cost-effective method may be to sequence the 16s rDNA gene from bacteria [33]. This gene acts like a molecular clock, determining which taxa the read is from just like a clock determines time, with different parts of the gene highlighting different granularity in the taxonomic tree [34]. This method and metagenomic methods return count data, where read depth is used as a measurement of how abundant that particular part of the microbiome is. Methods to determine absolute abundance, not just relative abundance, should be used as well—for example, spiking a known amount of a microbe or DNA into the sample. Once the count data has been determined, it may as well be batch corrected. From these data, microbes with known importance or microbial pathways with known biological relevance to the host may be determined using methods described below.

### 2.5. Mass Spectrometry for Biomolecules

Like for the nucleic acid methods, mass spectrometry (MS)-based methods share some similarity, but also have their unique properties. Each ome is first fractionated through liquid chromatography (LC), the parameters of which are determined according to its own unique biochemistry properties. In proteomics, the proteins are typically digested into shorter peptides first. After LC, proteins are sent through data dependent MS/MS or data independent acquisition. The software for calling peaks depends on the platform used, with the most popular being Skyline and Perseus [35,36]. Data-dependent acquisition generates more identified proteins but is less comparable between samples. Each method requires its own data analysis software to call peaks, for example, openSWATH for the data independent acquisition [37]. Furthermore, even with data from a single piece of software, the library compared against is absolutely essential for data quality. For example, the TWIN library may produce particularly good data with the openSWATH platform [38].

Metabolomics data can be generated in different platforms, such as reversed RPLC-MS (reversed phase liquid chromatography-mass spectrometry), HILIC-MS (hydrophilic interaction chromatography-mass spectrometry), and so on. These are then imported into Progenesis QI 2.3 software which is able to convert spectra data to data matrix for further downstream analysis. Further data preprocessing steps include but not limited to filtering noise signals, data imputation, retention time adjustment, and data normalization [39]. Removing batch effects is one of the most crucial tasks in metabolomics, and various classic and advanced methods have been developed. Recently, Feihn et al. published a random forest model-based normalization method SERRF (systematic error removal using random forest), and the authors claimed that this method outperforms other normalization methods, including median, PQN, linear method, and LOESS [40]. This normalization can be applied on both untargeted metabolomic and lipidomic datasets. After data cleaning, one can use either an in-house metabolite library or public databases (HMDB, Metlin, MassBank, NIST, etc.) for metabolite annotation. With different available data, the annotation needs to be defined clearly with confidence levels. Our laboratory uses a Lipidyzer, a semi-targeted lipidomics platform, to determine the lipid absolute abundance by using lipid chemical standards [41]. The software that calls the lipid species is LWM (Lipidomics Workflow Manager), although this area is a fertile one for growth.

From these methods, individual molecules, proteins, and lipid species may be associated with biological questions of interest. Furthermore, in the lipidomic data classes of molecules may be enriched from their individual species. For example, triacylglycerols as a whole, not just individual triglyceride species, may be associated with the biological question of interest. Proteomic and metabolomic data enrichments may be performed with DAVID or MetaboAnalyst [42]. Ingenuity pathway analysis (IPA) from Qiagen may also be used for enrichments of the proteomic and metabolomic data, and it may also be used to integrate the two together. The Kolmogorov–Smirnov method is an alternative approach for pathway/chemical class enrichment analysis in the metabolomics and lipidomics field, which is able to use ranked significance levels as input. There are many other pre-written computer programs available, such as IPA, to analyze multi-omics data (reviewed in [43]), but we will focus on methods for developing your own customized pipeline, rather than pre-built Web-based software.

From all these individual methods, information is gleaned about that particular omic measurement. Furthermore, these methods all generate data structured similarly that facilitate omic integration. They all generate a list of analytes for every sample, be it a transcriptome, microbiome, proteome, lipidome, or metabolome. These analytes are then associated with a particular intensity (Figure 1). There are many differences between these omic measurements, but this similarity in data structure facilitates downstream analysis (Figure 1). There are many other omics measurements that share similarities with those mentioned, and there are numerous databases containing already-generated datasets, which may also be used for integrative multiomics rather than generating new data [44]. For example, http://education.knoweng.org/sequenceng/ mentions 68 different next-generation sequencing technologies, most of which are omics measurements. Nonetheless, most share similarities with those already discussed here.

## 3. Designing a Quality Study

The first step in understanding an analysis flow for integrative multi-omics is determined by your study design. Cross-sectional and association studies are beneficial in their relative ease to implement, and their ability to generate large amounts of data (Figure 2a). Typically, cross-sectional studies do not involve a randomized intervention, precluding causal inference. They involve taking a population split between cases and controls, and then sampling them evenly, and are excellent methods for determining associations.

Conversely, longitudinal studies are relatively difficult to recruit large numbers of participants to because they generate large numbers of time points and become expensive. However, the longitudinal nature increases the statistical power of a relatively small number of participants [45] (Figure 2b). Longitudinal studies further facilitate making causal inferences and allow for more accurate predictions. Each study design, with its strengths and weaknesses, has a slightly different flow of analysis. Wherever possible, the multiple omic measurements should be selected not staggered in time. For example, if the treatment course is seven days, all the participants should be sampled on the same days during treatment. This will greatly facilitate the analysis methods.

Some advantages of longitudinal studies include the ability to associate events chronologically with particular interventions or exposures. They allow a study of change over time, or a delta measurement from baseline, as discussed below, which can be more powerful than studying a single point in time for the effects of an exposure or intervention. They also allow for establishing the chronological order of events, which is essential for establishing causation—again, something that is precluded in cross-sectional association studies. There are relatively few negative effects other than the difficulty of recruiting large numbers of participants, but they may also include loss of individuals over time, confounding results [46].

In each individual, there are apparent biases in the technologies and analytical methods, which limit insights into biology. Often signals from individual omes are difficult to label as accurate or relevant because the information does not connect to the broader context of the system. Multi-omic integration offers an opportunity to use orthogonal methods to measure the same molecular pathways and processes. Such methods partially mitigate the inherent false positives and false negative rates in the single omes, as finding the similarities and biological connections supports the truly biologically-relevant information [47].

## 4. Analysis Methods for Multi-Omic Integration

### 4.1. Dimensionality Reduction

The first step in an omics study is to reduce the dimensionality of your data so they can be visualized. In a metagenome, for example, there may be hundreds of microbial species. This means that every sample is a data point with hundreds of dimensions. Dimensionality reduction techniques will take the data and reduce them to fewer dimensions, often as few as two or three, that represent most of the variation in the data. Then it is easier to visualize and use statistics that require fewer dimensions.

The first dimensionality reduction technique invented is principle component analysis, which is a widely used unsupervised method. This method, though yielding valuable results, is not the most statistically precise because it assumes normally distributed data. Anyone who works in omics will testify that the data are never normally distributed, although transformations can make the data approximately normal. One superior method is non-metric multidimensional scaling. This method is iterative and nonparametric, avoiding problems with unusual distributions, and it handles zero-truncated data well—a phenomenon in which in some samples a particular analyte is undetectable and in others it exists at a high level. Another method, tSNE, is particularly well designed to separate well defined groups. Besides t-SNE, UMAP (uniform manifold approximation and projection) [48] is a newly developed dimension reduction technique for non-linear relations. It usually implements faster than t-SNE, especially when it concerns large number of data points or a number of embedding dimensions greater than 2 or 3. It has many applications in single-cell sequencing data. Other methods include principal coordinate analysis and multidimensional scaling. Every method is capable of providing useful information; however, properly selecting a method can increase your statistical power.

The information gleaned from dimensionality reduction is similar across omic techniques. It can discover batch effects, particularly in mass spectral data. If two batches do not overlap, then additional correction techniques need to be applied. This method can find samples that failed, which would be represented as outliers in the data. Once data quality has been established, these methods can find any structure in the data that might be associated with biologically relevant variants. This is the most basic example where a metadatum, participant ID, may be grouped together. However, there are many more—sex, insulin resistant status, etc. In the case of the microbiome, it can also be used to measure beta diversity, as outlying samples will have different microbial compositions than the rest of the cohort.

### 4.2. Normalizing the Data

Once the structure of the data has been determined, omics measurements can be grouped together for integration. Usually they are done so after log, log2, or other transformations to facilitate downstream statistics [49,50]. The log transformation is normally used to make highly skewed data approximately approach a normal distribution. This can be useful both for facilitating the data to meet the assumption of statistic models and for making patterns in the data more interpretable. Microbiome data are so unusually distributed, other transformations may be applied, such as arcsin. With certain longitudinal designs they can be normalized to the baseline measurements to only measure the deltas from the baseline, reducing the effects of inter-individual variability. This is absolutely essential in longitudinal data to reduce the effects multiple individuals would have on biasing a sample, and is one of several strengths of that study design.

A z-score is another normalization method that standardizes all the analytes to the same range. This alleviates the problem of vastly different expression levels, facilitating grouping several different omes together for integration. For example, if one wanted to integrate the metabolome and gut microbiome, the values for the metabolome may be in the tens of millions, while analytes in the microbiome may be zero truncated, with most values being 0. To compare these two, particularly visually, they must be on a similar scale. Z-scoring makes the average value for every analyte 0, and then one standard deviation above that 1, etc.

### 4.3. Correlation Networks Analyses

Once these normalizations and transformations are performed, correlation metrics can inform one about the most basic relationships between the analytes. Pearson correlation coefficients (PCC) and spearman correlation coefficients (SCC) are the two most typical types of correlation metrics. The PCC is a parametric metric with more accuracy, whereas the SCC is more robust if outlier samples are present. One should target analytes of the most interest (e.g., only the significant molecules) if possible, because with too many analytes in networks, it is difficult to capture the most useful biological information and it is inclined to be masked by the underlying noise. Correlation networks are much more effective when dealing with deltas in longitudinal data that reduce interindividual variability. If more than one sample, not corrected to baseline, is from a single individual, such an analysis will be overfit and produce false positives. Additionally, one must always correct for multiple hypotheses during these projects to reduce false positives. In these data a Benjamini–Hochberg correction is appropriate. One may also use a Bonferroni correction, but in some omic studies that may overcorrect, losing true positives. Both will have their uses and may be differentially used in longitudinal baseline normalized vs. unnormalized data. This correlation analysis can be plotted as a network diagram, which is a fantastic visualization tool for this type of data. Though high-level visualizations, network diagrams offer compelling, informative overviews of interactions in biological systems [51].

When comparing interaction networks across different conditions, disease states, or interactions, a network analysis may provide you appropriate information about how the two states differ. A network analysis will provide one with total nodes (analytes) that are connected in the network, the total number total edges between the networks (significant correlations), and many other important relationships, such as the numbers of positive and negative correlations. Complementing visualizations, these summary statistics provide an excellent overall view of the co-correlations occurring in any multi-omic project. This type of topological analysis is not only able to provide practitioners straightforward and clear ideas when comparing multiple networks, but also provide insights into network hubs and centers, which may have many applications in drug target selection and identification of key regulators. There are several packages for R—igraph, statnet, ggnetwork, ggnet, ggraph, etc.—with highly related functionalities that perform these analyses [52,53]. R packages “igraph” and “statnet” are able to provide quick visualizations, which are good for a quick exploration about the network structure but are not necessarily the most efficient ways for aesthetically perfect visualization. R packages “ggnet” and “ggnetwork” are very similar packages, and both seem to use a variant of the ggplot syntax, meaning that they would be advantageous if you are familiar with the ggplot system.

### 4.4. Cross-Sectional Analyses and Testing Categorical Variables

In a cross-sectional study, when testing a single analyte between two sets of samples, the nonparametric version of the student’s *t*-test, Wilcoxon rank sum test, is appropriate. A *t*-test assumes a normal-like distribution and should be used with care, as omics measurements are extremely rarely Gaussian. If confident that prior information will be obtained before the test, Bayesian counterparts to these tests will provide more power. These are not necessary, and should only be used by an expert. In a cross-sectional study where two categories are being tested against, one may further use logistic regression as a means of regressing between these categories. This regression fits a curve to binary data, generating an odds ratio and *p*-value.

When analyzing across more than two categories of data, one should use the non-parametric analysis of variance (ANOVA), Kruskal–Wallis. This method may be used to test a trend in your data over categorical variables. When correcting for multiple variables, one may use a multivariate ANOVA (MANOVA), but this should be used with care because ANOVAs assume a normal distribution. To avoid these assumptions about distributions, a permutational multivariate analysis of variance (PERMANOVA) should be used.

### 4.5. Testing along Continuous Variables

Another method determining trends over categorical variables is multiple linear regression. Like ANOVA, this may be used to find trends in one or more categorical variables. However, multiple linear regression can find trends over continuous variables as well, or any combination thereof. Although multiple linear regression also assumes a normal distribution, it can still be a valuable tool for detecting trends in data and is widely used by multi-omic researchers. In cases like this, where the statistical assumptions do not perfectly match data distributions, orthologous methods should be used for confident assessments.

Even more sophisticated than multiple linear regression is a mixed model. These can find trends in data and can also find the variance in data for random variables. Random variables are those that are randomly distributed in your data, say a random assignment of sex, so they are not associated with the outcome variable. Nonetheless, these variables can add variance, making the data noisier. Further, these mixed models can select other distributions than Gaussian, such as Poisson, so variables that violate normality may be modeled better. Mixed models are appropriate to account for complicated and heterogeneous datasets with confounders—gender, race, age, BMI, etc. Mixed models are particularly well suited for tracking longitudinal data [45]. Together, these methods are powerful for detecting trends in the data.

### 4.6. Clustering Algorithms

Clustering algorithms group similar samples or analytes together. Two primary clustering algorithms are hierarchical and k-means clustering. These are “hard” clustering algorithms which force analytes or samples into particular groups. This may be useful to determine whether samples cluster by individual, batch, or some other biological measurement, for dimensionality reduction techniques. They can also be used to determine outliers in the data, which may be of special interest to the researcher.

To find trends in longitudinal data, fuzzy c-means clustering is a powerful tool. The R mfuzz package provides tools for this analysis [54]. This is a “soft” clustering algorithm, giving analytes a score known as membership in every cluster, rather than forcing them into a single cluster. However, like other clustering algorithms, it still finds analytes with similar expression profiles. Using the previously mentioned z-scores, c-means clustering finds longitudinal trends in data for multiple omic measurements. These trends are powerful if one wants to find dose, temporal, delayed, or other response patterns in multi-omics data [55,56].

One of the most critical and haunting issues in clustering is to determine the optimal number of clusters. Selecting an inappropriately small number of clusters would cause the missing detection of some meaningful molecular trends and clusters, whereas an improperly large number of clusters may result in redundancy of cluster detection. There are several ways to assist the selection of the optimal number of clusters. One of the classic methods is called the elbow method, which calculates the within-cluster sum of squared error (wss). This method is widely applied; however, it gets tricky to determine the “elbow” point. Another way to survey this issue is to calculate minimal centroid distance, which is similar to the elbow method, aiming to find the “elbow” point to gain the minimal centroid distance. Another more efficient method is to calculate the correlations between cluster centroids, and decide on the optimal number of clusters once high positive correlations (e.g., 0.85) are detected.

Another method of clustering, supervised clustering, involves placing a priori information into a model before using the clustering algorithm. For example, if you have cases and controls, these may be entered into the data beforehand, or if you have longitudinal data with doses, the baseline controls may be contrasted with the doses. Categorical variables are required for this type of clustering, but they are an excellent method of assuring one will find analytes with similar expressions in the data [46].

### 4.7. Feature Selection for Covarying Analytes

A powerful tool in the arsenal of multi-omics researchers is feature selection. In some data, the analytes strongly covary. For example, in the metagenome, if one organism increases it will have an effect on every other organism in the system. In such circumstances it may be difficult to know which of these analytes to prioritize putting in a model. Least absolute shrinkage and selection operator (LASSO) and ridge regression tackle these problems. These functions will weight or eliminate the variables with the most and least explanatory power in your model. This way, future analyses may be performed on more manageable and more meaningful data, which may also increase statistical power. There are numerous feature selection methods, and descriptions and comparisons of all of them are obviously beyond the scope of our review. We mainly highlight two of them (LASSO and ridge) because they are widely applied penalized algorithms that reduce model complexity and prevent over-fitting which may result from simple linear regression. The main principle of these two regularization methods is to restrict or shrink the coefficients towards zero for the non-impactful features, in order to reach the goal of feature selection.

### 4.8. Machine Learning

Machine learning is an important subset of artificial intelligence, and nowadays has drawn attention in various fields. In omics studies, machine learning is widely applied on classification and prediction problems by using omics profiling data. Different suites of machine learning algorithms are suitable for classification and prediction scientific problems. Classification and prediction, as two main branches of machine learning, depend on the types of tasks or problems that are intended to be solved by machine learning and are either categorical (classification) or continuous (prediction). There are three main types of machine learning algorithms: unsupervised, supervised, and reinforcement learning.

Classification and regression are the two main prediction domains in the machine learning field. Classification is the problem of predicting a discrete class output, while regression is to predict a continuous quantity output. Due to the pronounced differences in principles for these two domains, the modeling algorithms applied on these two problems are different. Some algorithms can be used for both with minor modifications, e.g., decision trees and artificial neural networks, whereas some algorithms are only suitable for either classification or regression problems—e.g., logistic regression can only be used for classification, and linear regression is only for regression predictive modeling. More importantly, the matrices that are used to evaluate models varies for classification, e.g., accuracy, are usually used for assessing classification models but not regression algorithms, whereas root mean squared error (RMSE) is only for regression predictive models but not classification models.

One of the useful supervised machine learning algorithms in multi-omics is the random forest. A random forest is not a black box telling you which parameters are the most predictive of biology. Conversely, neural networks and deep learning are typically not appropriate for multi-omics datasets because of the structure of the multi-omics data, normally with more variables than sample size. Neural networks provide more accurate predictions when there are many samples, and relatively few measurements per sample. Multi-omic studies are typically the opposite, with relatively few samples but many, many measurements per sample. There is nothing in principle preventing neural networks from working on multi-omic datasets, but rather the practical considerations of how these studies are designed. Further, neural networks and deep learning are “black boxes” where the decisions of the algorithm are unknown to the researcher. For these reasons, random forests may provide better predictions in multiomis data, as measured by recall, area under receiver operating receiver curve, and the Mathews correlation coefficient [57]. Though more sophisticated than other analysis methods, these machine learning techniques are phenomenal for first, exploratory, unbiased passes on the data. They will determine which features are most predictive of data outcome, and what to look for as grounding during other analyses.

## 5. Conclusions

There are several limitations in multi-omic integration, including potential statistical overfitting, varying distributions between analytes, and limitations in throughput for some techniques [43,58,59]. Nonetheless, multi-omics are a suite of tools that allow researchers to answer questions with unparalleled depth. These measurements are not perfect in themselves, and consistency between omic measurements will ensure the discoveries are true to the underlying biological reality. Furthermore, there are no perfect methods for analyzing these data. A researcher should be confident in their findings when their discovery comes up in multiple omes but also when discovered through multiple analysis and statistical methods. What we have discussed here is not exhaustive of the excellent analysis methods that exist, but we are confident that any researcher employing these techniques will find the trends present in their multi-omic dataset successfully.

## Figures and Tables

**Figure 1 biomolecules-10-01606-f001:**
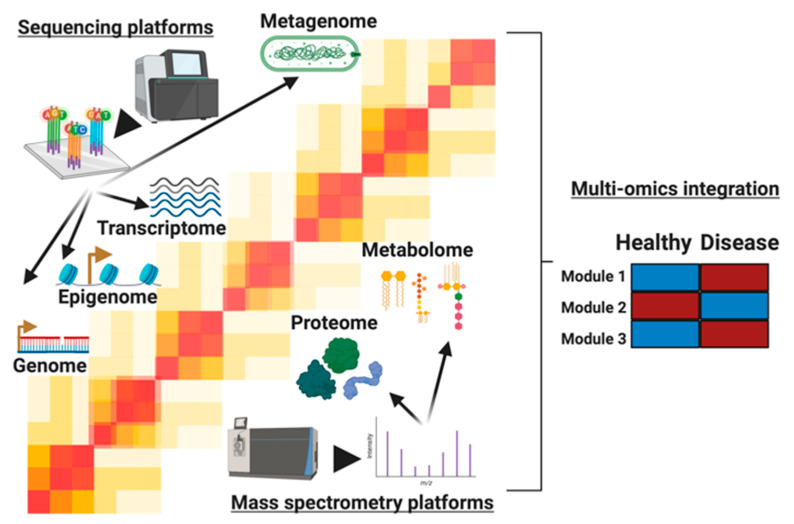
The molecules profiled in multi-omics studies. We describe 6 levels of information, starting from the bottom to the top: genome, epigenome, transcriptome, proteome, metabolome, and metagenome. The genome, epigenome, transcriptome, and metagenome are profiled by sequencing-based technologies such as sequencing by synthesis, depicted here, to profile a comprehensive set of nucleic acid molecules. On the other hand, mass spectrometers generate proteome and metabolome profiles as depicted here through measurements of biomolecules’ masses and charges. For overlapping technologies, each omic level provides unique information and insights into cellular activity present in conditions being studied. By leveraging the layers of information, longitudinal and cross-sectional multi-omics studies find modules (e.g., cell signaling pathways) that are differential between healthy and disease states. These modules represent complex system biology networks that give precise insights into the molecular dysregulation in disease states.

**Figure 2 biomolecules-10-01606-f002:**
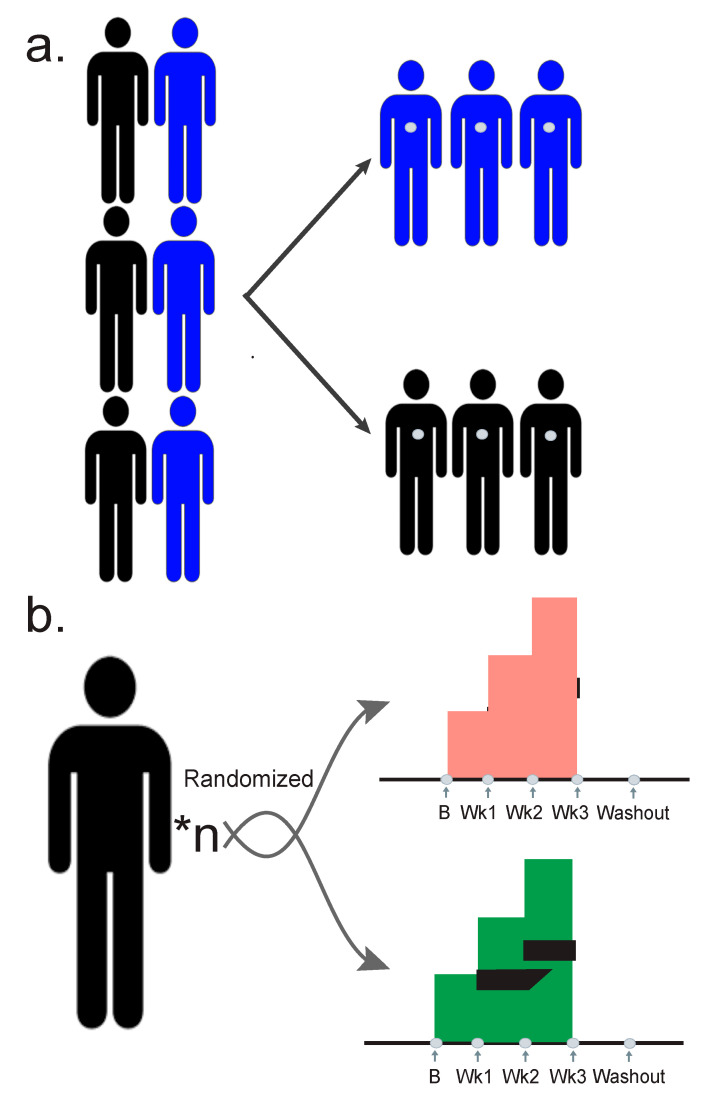
Typical multiomic study designs. Gray dots represent samples taken. (**a**) A case control observational study. A population is taken with participants that have the phenotype of interest (cases) and those without (controls). Cases and controls are sampled in even amounts. (**b**) A randomized longitudinal study where n participants are randomized into two arms of a study. In this case an increasing treatment dose is administered, and samples are taken every week.

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
