# Peer review of "A Customizable Analysis Flow in Integrative Multi-Omics"

_biomolecules, 2020, doi:10.3390/biom10121606_

Round 1

Reviewer 1 Report

This article reviewers various components of a multiomics data integration pipeline. The authors have substantial previous experience with this kind of analysis and this experience has enabled them to provide a detailed review. Overall, the article is well-written and a is a good fit for the journal. My specific comments are provided below.

L: Line number

MAJOR:

L276: The authors should consider removing section 4.1 all together. These generalizations are not accurate and highly dependent on the specific libraries used. Packages in R often rely on low-level computations in C++ or Fortran (as just one example) that are faster than native implementations in Python.

L448: Stating that one algorithm is “better” that the other should be reserved for when empirical data exists. For purposes of this article I think it would be sufficient to discuss perceived strengths and weaknesses and leave the rest to actual benchmarking studies.

MINOR:

L217: Remove one “available”

L221: facilitates -> facilitate

L254: This is an important point. Ultimately, causality is inferred through interventions. Longitudinal studies are of course important and how they could be used in a multiomics setting should be further expanded upon. Linear mixed effect models would be particularly important.

L256: We could not understand this sentence. Please rephrase.

L273: This seems out of place. Perhaps included in the text by mistake?

L285: While dimension reduction is broadly used for visualization, it can also be used for other purposes (e.g. feature selection). These should be discussed as well.

L289: The number of output dimensions does not have to be limited to 2 or 3.

L291: principle -> principal

L295: The data does not need to be perfectly normal. Only approximately.

L313-315: Please add citations as appropriate.

L394-402: This discussion likely belongs in 4.3. Normalizing the data.

L432-437: This is an important point. The authors should consider rewriting this section to focus it on classification and regression (as opposed to prediction).

Nima Aghaeepour

(Voluntary disclosure of reviewer’s name)

Reviewer 2 Report

In this manuscript, the authors review the methods of analysis of multi-omics data. Although, the topic sound very interesting, I was disappointed after reading the article. In abstract, the authors stated that they use 6 omic measurement, however in text I haven’t found any detailed description how to perform the integrative analysis using all of them, what methods could be used and what are their advantages and drawbacks. In my opinion, the authors tried to cover too many different and complex topics here, which resulted in a vague description of only most commonly used methods. In many aspects, they propose to use some methods or thresholds, without detailed explanation why. Also, I found many inaccuracies in the text. At last, it was very surprising that chapters 2 and 3 that cover single omics analysis and design of the study, cite around 40 papers, while chapter 4 that covers multi omics integration, which should be a key aspect of this work, cite only 3 papers. Therefore, I suggest rejection of the manuscript.

Major comments:

  • „This task is extraordinarily complicated because of the numerous mRNA isoforms, and the non-normal distribution of reads” – should we afraid analysis of non-normally distributed data? There are multiple non-parametric methods or models based on other than normal distributions, e.g. negative binomial. Ultimately, there are methods to normalize/transform the data to get closer to normal.
  • „Generally speaking a PHRED score of 30 is used as a threshold for keeping a read or not” – how can we state that based on only single publication? In other, you can find the different threshold values used in filtering.
  • Line 63: Multi-qc software only integrates results of other methods, so it cannot be used to calculate PHRED scores.
  • “From these files variants between the sequenced genome and referenced genome can be determined using Samtools or other software saved as a variance call format (VCF) file” – how can a reader know what do you mean by variants?
  • Why do your recommend to use TPM or RPKM? What about VST normalized data?
  • “The program sva::COMBAT() from R is excellently suited for batch correction [25], when not employing software like DESeq that performs normalization” – normalization and batch correction are two different concepts. DESeq does not remove batch effect, but the can control it by introduction batch info in the design of the experiment.
  • Line 151: You describe the problem of deconvolution of transcriptomics data, but only 1 method is shown. What about CIBERSORT or others?
  • Enrichment analysis methods are described twice in text, namely in chapter 2.3 and 2.5. Could you explain why?
  • Figure 2 takes whole page. Is it necessary?
  • Chapter 4.1: Why only 2 programming languages are mentioned?
  • Chapter 4.2: UMAP is a new dimensionality reduction/data visualization method that becomes the new standard in the analysis, however it’s not mentioned here.
  • Line 314: “Usually they are done so after log, log2, or other transformations to facilitate downstream statistics.” – It’s good to describe why and when data transformations are necessary.
  • Chapter 4.7: Currently, there are hundreds or thousands of feature selection methods, but here only Lasso and ridge regression are mentioned. Why these two?
  • There are multiple places in the text describing different methods and concepts, but the citations are missing.

Minor comments:

  • Multi omics or multi-omics?
  • Please define shortcuts at their first usage, e.g. ATAC seq, SNPs, etc.
  • Remove double spaces from the text.

Reviewer 3 Report

The manuscript by Lancaster et al., titled “A Customizable Analysis Flow in Integrative Multi Omics” describes a review focusing on how to analyze and integrate multiOMICS datasets. The article is well written with a good background on the basics of OMICS as well as data analysis platforms. The authors have written it in a way that suits extremely well for biologists with no background in data analysis. The authors have explained various state-of-the-art technologies, data analysis platforms, reducing the data complexity as well as basic statistics at sufficient depth to cater the needs of biologists and data scientists. I strongly feel that this article will be of tremendous use for the biomedical research community. I recommend the acceptance of the article for publication.

A few minor points:

  1. It would be nice if the authors can highlight a few published studies that illustrates the application of some of the concepts discussed here in the context of addressing biological research questions.
  2. The authors can also pinpoint a few public repositories where readers can download some of these OMICS datasets so that they can try out some of the data analysis pipelines mentioned in the article.
